# Inequality in the Distribution and Utilization of Healthcare Resources in Kazakhstan (2002–2023): A Spatiotemporal Analysis

**DOI:** 10.3390/ijerph22111762

**Published:** 2025-11-20

**Authors:** Askhat Shaltynov, Madina Abenova, Assel Baibussinova, Yuliya Semenova, Nazarbek Omarov, Gulnaz Tanatarova, Aigerim Sepbossynova, Jorge Rocha

**Affiliations:** 1Epidemiology and Biostatistics Department, Semey Medical University, Semey 071400, Kazakhstan; askhat.shaltynov@smu.edu.kz (A.S.); madina.abenova@smu.edu.kz (M.A.); assel.baibussinova@smu.edu.kz (A.B.); 2Department of Medicine, Nazarbayev University, Astana 010000, Kazakhstan; 3Department of Surgery, Semey Medical University, Semey 071400, Kazakhstan; nazarbek.omarov@smu.edu.kz; 4Department of Healthcare of Abay Region, Semey 071400, Kazakhstan; 5Independent Researcher, Semey 071400, Kazakhstan; sepbossynova@gmail.com; 6Centre of Geographical Studies, Institute of Geography and Spatial Planning, University of Lisbon, 1649-004 Lisboa, Portugal; jorge.rocha@edu.ulisboa.pt

**Keywords:** healthcare inequality, universal health coverage, Kazakhstan, spatiotemporal analysis, Gini coefficient, concentration index

## Abstract

Global progress toward achieving universal health coverage (UHC) by 2030 remains insufficient, as significant regional disparities in access to healthcare persist. In Kazakhstan, the uneven distribution of healthcare resources continues despite reforms aimed at improving equity. This retrospective study analyzed inequalities in the distribution of medical infrastructure, healthcare personnel, and service utilization across 14 regions and 2 cities of republican significance from 2002 to 2023. Data were obtained from national statistical reports on healthcare and population income. The analysis included the following indicators: the number of primary care facilities, hospital beds, healthcare personnel, outpatient visits, and hospitalizations per population. Inequality was assessed using the Gini coefficient and the concentration index, and spatiotemporal trends were visualized through cluster analysis in ArcGIS PRO. Results revealed that southern and western regions exhibit lower availability of hospital beds and healthcare personnel, with moderate levels of inequality particularly evident in outpatient care. Despite Kazakhstan’s commitment to UHC, these disparities underscore the need for further measures to ensure equitable access to healthcare services.

## 1. Introduction

The world is lagging behind in making significant progress towards achieving universal healthcare coverage (SDG Target 3.8) by 2030. This global trend is consistent across all regions and the majority of countries. This issue persists despite advancements in healthcare accessibility and coverage at the national level, as disparities persist within various regions or demographic groups. These within-country variations in healthcare access and utilization underscore the ongoing challenge of addressing inequality within healthcare systems [1].

Social and economic factors, such as GDP, GDP per capita healthcare expenditures, unemployment, poverty, and population size, influence the progress of universal healthcare coverage regardless of the maturity of the system or geographical characteristics [2,3].

The Republic of Kazakhstan (hereafter referred to as Kazakhstan) is an upper-middle-income country located in Central Asia. It gained independence in 1991 following the dissolution of the Soviet Union. Kazakhstan’s healthcare system originates from the Semashko model and retains many of its features to this day. One of these features is an over-reliance on hospital care, characterized by an extensive network of hospitals and high hospitalization rates [4]. Despite reductions over the 2000s, hospitalization rates in Kazakhstan remain high compared to developed countries. In 2011, the average hospital stay was 9.9 days, with a hospitalization ratio of 16.3 per 100 people and a hospital bed capacity of 100 per 10,000 inhabitants [5].

Kazakhstan is committed to achieving universal healthcare coverage, as reflected in the Healthcare Development Concept of the Republic of Kazakhstan until 2026. This concept aims to transition from disparities in medical services between urban and rural areas to the widespread provision of quality healthcare services [6]. Since 1 January 2020, Kazakhstan has operated a compulsory Social Health Insurance (SHI) system alongside the State Guaranteed Benefits Package, providing healthcare coverage for all citizens and permanent residents as part of progress toward UHC [7].

Nevertheless, over the past three decades, there has consistently been a notable disparity in the economic well-being of individuals across regions of Kazakhstan, particularly where the country’s leadership has embarked on reforms aimed at cultivating a middle class. Significant disparities in health indicators have been identified, with poor health outcomes largely attributed to living conditions and income levels [8,9].

Income levels significantly affect the concentration index(CI), as higher income groups often have better access to healthcare services and higher educational attainment is associated with better health outcomes and greater healthcare utilization, contributing to the CI [10,11]. Utilization of healthcare services is often pro-rich, meaning wealthier individuals use more healthcare services [11,12]. This is evident in both outpatient and inpatient services. Regional economic disparities also play a role, with wealthier regions having better healthcare infrastructure and services [11]. This directly relates to healthcare utilization, which is often pro-rich, meaning wealthier individuals use more healthcare services. This is evident in both outpatient and inpatient services [12].

Health service utilization is shaped by both demand-side and supply-side determinants. Following the Andersen Behavioral Model of Health Services Use, socioeconomic context (for example, income) influences an individual’s ability to seek care, while predisposing and enabling factors determine whether care is actually obtained [13]. At the same time, in line with the WHO health system perspective, the distribution of infrastructure (primary care facilities, hospital beds), workforce (medical personnel), and financial protection mechanisms are system-level inputs that determine service availability and capacity at the regional level [14]. Inequality in utilization can therefore be interpreted as the combined result of unequal resource allocation, uneven system capacity, and differences in local population needs.

Geographic Information System (GIS) based time series cluster analysis has emerged as a powerful tool for examining complex spatial-temporal patterns in healthcare—ranging from disease incidence to resource distribution—and informing effective, data-driven public health interventions [15]. Despite the proven value of these methods, their application in Kazakhstan remains largely unexplored, signaling a pressing need to adopt advanced analytic techniques for reducing health disparities and strengthening healthcare infrastructure [16]. While spatiotemporal clustering methods have been increasingly applied in public health internationally, and standardized approaches to measuring health inequality are well established, there remains no long-term, country-wide analysis that combines these methods for Kazakhstan or the Central Asian region. The present study addresses this gap by integrating inequality indices with spatiotemporal clustering over 2002–2023 [17,18].

The aim of this study was to assess regional inequality in the distribution and utilization of healthcare resources across Kazakhstan during 2002–2023, using Gini and concentration indices and spatiotemporal cluster analysis to identify patterns of disparity among regions.

## 2. Materials and Methods

### 2.1. Study Design and Area

Retrospective analysis of inequality in the distribution of healthcare resources and utilization of medical services among regions of Kazakhstan. With a territory spanning 2724.9 thousand square kilometers, the country ranks ninth in the world by area. It shares borders with Russia to the north and west—spanning 7591 km (the longest continuous land border in the world), China to the east—1783 km, Kyrgyzstan to the south—1242 km, Uzbekistan—2351 km, and Turkmenistan—426 km [19]. The total length of land borders is 13,200 km. The administrative-territorial structure of the country comprises 17 regions and 3 cities of republican significance. However, we used the 2017 administrative-territorial division with 14 regions and 2 republican cities for clustering, as it was in place for most of the study period (Figure 1).

### 2.2. Data Sources

This study is based on secondary aggregated data extracted from statistical compilations titled ‘Population Health of the Republic of Kazakhstan and Activities of Healthcare Organizations’ for the years 2002–2023 [20]. These compilations are the primary government-approved sources for healthcare statistics, published by authorized state bodies and widely used for policy analysis and public health research. Data on per capita nominal cash incomes of the population by region were sourced from materials of the National Statistics Bureau of the Agency for Strategic Planning and Reforms of the Republic of Kazakhstan for the same period [21]. As the country’s central statistical authority, the Bureau ensures the accuracy and consistency of socioeconomic indicators, making these datasets a reliable basis for analysis. However, it should be noted that income data from the Bureau are only available from 2015 onward.

### 2.3. Variables

We selected indicators reflecting healthcare infrastructure, utilization, and economic disparities. These include the number of primary care institutions per 10,000 population, hospital beds per 10,000 population, outpatient visits per person per year, hospitalizations per 100 residents, physicians per 10,000 population. Additionally, we included per capita income in Kazakhstani Tenge (KZT), approximately 0.0020 USD per KZT, as a ranking variable for calculating the concentration index, along with the Gini index to assess socioeconomic inequalities in healthcare access. Population data used for standardization were obtained from the same official statistical compilation, ‘Population Health of the Republic of Kazakhstan and Activities of Healthcare Organizations.’ Raw indicators of healthcare facilities were standardized per 10,000 population for comparability. For newly established regions and the city of Shymkent, data for 2022 were presented as the average values of the newly formed region and its original region. For example, the values for East Kazakhstan Region in 2022 were calculated as the average of East Kazakhstan Region and Abai Region, while the values for South Kazakhstan Region were represented as the average of Shymkent and Turkestan Region.

### 2.4. Inequality Indices Measurement

The CI is used to measure the degree of inequality in the distribution of a variable, often in the context of health-related measures by income. It indicates the share of the total amount of a variable that needs redistributing to achieve a concentration index of zero. This index has been applied to various fields, including the analysis of socioeconomic inequality in health. The CI equals to the absolute value of twice the area between the equality line and the concentration curve, allowing the calculation of perceived poor and very poor health using Equation (1):(1)$$CI=\frac{2}{\mu }cov\left(y_{i},R_{i}\right)$$
where *R_i_*—is the rank of the individual in the income distribution; *μ* = $\frac{1}{n}\sum_{i=1}^{n}h_{i}$, where *n*—*n* is the number of respondents in the sample.

The concentration index ranges from −1.0 (a situation where the poorest household contributes all health-care payments) to +1.0 (where all health-care payments are made by the richest household).

On the other hand, the Gini coefficient is a widely used measure of inequality, typically applied to income or wealth distributions but also adaptable to other contexts. It is defined as the area between the Lorenz curve of a distribution and the line of equality, normalized to be between zero and one. The Gini coefficient can be used to evaluate inequality in various distributions, including those with non-economic data, such as heart rate variability. However, it is important to note that the Gini coefficient requires all samples to be positive, and extensions have been proposed to handle cases where the sum of samples is less than zero. The Gini coefficient is calculated using the Brown Equation (2):(2)$$G=1-\sum_{k=2}^{n}\left(X_{k}-X_{k-1}\right)\left(Y_{k}+Y_{k-1}\right)\vee$$

The value of the Gini coefficient ranges from 0 to 1. A coefficient of 0 indicates perfect equality. A coefficient of 1 indicates perfect inequality.

The Gini coefficient is suitable for measuring inequality in independent variables, especially when dealing with positive data. It is adaptable to various contexts beyond income and wealth. On the other hand, the Concentration Index is effective for dependent variables, particularly in health-related studies, and provides a clear measure of the redistribution needed to achieve equality.

### 2.5. Statistical Analysis

The statistical analysis was conducted using STATA version 13 in conindex: Estimation of concentration indices package [22]. ArcGIS PRO 3.2. Time Series Clustering tool used to demonstrate spatiotemporal distribution of healthcare resources and utilization across the regions of Kazakhstan from 2002 to 2023. The Time Series Clustering tool [23] identifies similar locations within the space-time cube divides them into distinct clusters, where each cluster contains similar time series characteristics [24]. The tool generates a class of 2D objects, displaying each location in the cube, marked with its cluster membership, as well as informational messages and an output table for charts. These charts display representative time series for each cluster and time series for each location in the space-time cube [25]. The average annual percentage change and relative increase were calculated for Gini index trends using WinPepi version 11.65, a statistical software designed for epidemiological analysis [26]. The average annual percentage change measures the average rate of change in an indicator over a specified period and is particularly useful for identifying long-term trends [27]. Relative Increase quantifies the proportional change in a variable over time providing insights into the growth or decline of the variable. To assess the presence of statistically significant trends in the time series data we applied the Mann–Kendall trend test, which was performed automatically using the time-series clustering tool in ArcGIS. The Mann–Kendall test is a non-parametric method used to detect monotonic trends in a time series without requiring data to follow a specific distribution and helps determine whether an observed trend is statistically significant or a result of random variation [28].

## 3. Results

Table 1 presents general information about the number of primary care institutions per 10,000 population, beds per 10,000 population, outpatient visits per person per year, hospitalizations per 100 population, physicians (except dentists) per 10,000 and per capita nominal income in KZT of the population across the regions of Kazakhstan in 2023. In 2023, physician density ranged from 26.3 per 10,000 population in Akmola to 48.7 in Aktobe, while hospital-bed supply was lowest in Mangystau (30.7) and highest in North Kazakhstan (62.2). Regions with higher income, such as Atyrau, Astana, and Almaty, consistently showed greater healthcare resource availability, illustrating a persistent north–south gradient.

In 2023, regional disparities were non-trivial. Physician and hospital-bed availability was systematically higher in northern industrialized regions, whereas southern regions combined lower availability with higher service use. This pattern foreshadows the spatiotemporal clusters described below and motivates the inequality analysis.

Figure 2 illustrates three spatiotemporal clusters in the distribution of Primary Health Care (PHC) facilities across Kazakhstan’s regions. Despite a temporary increase to around 2 facilities per 10,000 population in 2013, the lowest long-term values remain concentrated in South Kazakhstan, Mangystau, and the city of Astana. The Mann–Kendall test indicates statistically significant decreasing trends for all clusters (Z = −3.56; Z = −4.57; Z = −2.31; *p* ≤ 0.02), confirming a gradual decline in PHC facility density. The overall pattern suggests a persistent north–south gradient: regions that started from lower PHC density did not converge toward better-performing areas during 2002–2023. These findings imply that the expansion of PHC infrastructure has slowed nationwide, with resource allocation remaining uneven and largely path-dependent.

According to Figure 3, hospital bed availability per 10,000 population is consistently lower in clusters located in the southern and western parts of Kazakhstan, including Almaty, Jambyl, South Kazakhstan, Aktobe, and Mangystau regions. The Mann–Kendall test shows a significant negative trend for all clusters (Z = −4.40; Z = −5.41; Z = −4.74; all *p* < 0.001), confirming a long-term decline in bed density. The temporal profiles of all clusters share a similar trajectory: a gradual decrease until 2019, followed by a short-term surge in 2020–2021 associated with the emergency expansion of hospital capacity during the COVID-19 pandemic, and a subsequent partial return toward pre-pandemic levels by 2023. Overall, these results suggest that Kazakhstan’s hospital capacity responded temporarily to crisis demands but the long-term trend remains downward. 

Figure 4 presents the spatial and temporal patterns in the availability of medical personnel (excluding dentists per 10,000 population). The cities of Astana and Almaty consistently maintain nearly twice as many healthcare staff as other regions throughout the study period, reflecting the concentration of National and Republican medical centers that provide multidisciplinary care at the national level. The red cluster, comprising Aktobe, Karaganda, Pavlodar, and East Kazakhstan Regions, also shows higher staffing levels due to the presence of medical universities and university-affiliated hospitals in these regions. The Mann–Kendall test indicates heterogeneous trends: a significant increasing trend in Cluster 1 (Z = +5.02, *p* < 0.001), a significant decreasing trend in Cluster 2 (Z = −3.59, *p* < 0.001), and a stable trend in Cluster 3 (Z = +0.90, *p* = 0.37). These results reflect that while certain educational and metropolitan regions have strengthened their human resource base, other areas continue to experience gradual declines.

Figure 5 shows regional disparities and time trends in outpatient care utilization, measured as visits per resident per year. Until 2018, Mangystau Region had the highest visit rates, but by 2021–2023 its values dropped below the national average. The Mann–Kendall test reveals significant negative trends across all clusters (Z = −4.10; Z = −3.47; Z = −4.88; all *p* < 0.001), confirming a gradual nationwide decline in outpatient activity. Regions in the north, east, and south, together with Atyrau Region, form clusters with low utilization, indicating persistent inequality in access to primary-level services. The overall downward trend may reflect shifts in health-seeking behavior and service organization during and after the COVID-19 pandemic.

Figure 6 illustrates spatiotemporal clusters of hospitalizations per 100 population in Kazakhstan. The green cluster, representing Astana, shows a pronounced increase up to 2013, associated with the expansion of National Medical Centers that serve the entire country. In contrast, other clusters demonstrate decreasing trends (Z = −3.40; Z = −4.43; *p* < 0.001), consistent with national policies to reduce inpatient volumes and strengthen primary care. A temporary drop in 2020 followed by a rise in 2021 likely reflects deferred hospital demand after the COVID-19 pandemic. The red clusters, covering less populated northern regions, exhibit persistently high hospitalization rates but low outpatient utilization, suggesting a supply-driven pattern of inpatient care. Overall, these trends reveal a gradual structural shift from hospital-based to outpatient-oriented service delivery, while capacity and utilization remain uneven across regions.

The aggregated Gini indices for primary care institutions per 10,000, hospital beds per 10,000 and physicians per 10,000 had high levels of equity (*p* < 0.001) in 2023 (Table 2).

As for utilization, then concentration index for outpatient visits ranked by average per capita nominal income of the population indicates slightly higher utilization among poor regions and there was proportional scheme for inpatient hospitalizations in 2023 (Table 3).

In Figure 7, we observe that the Gini indices for inequality in healthcare resource availability between regions have remained below 0.22 throughout the observation period. Since 2016, there has been a gradual decline in the Gini index for the availability of medical personnel, indicating a more even distribution of healthcare professionals across regions. However, inequality in the distribution of hospital beds has increased since 2019, reaching 0.14 in 2021. The negative peak of the Gini index in 2013 was due to a sharp increase in the number of primary healthcare organizations in the South Kazakhstan region, primarily driven by institutions outside the jurisdiction of the Ministry of Health.

Regarding the concentration indices, which assess inequality in the utilization of medical services relative to income levels, they suggest a generally proportional system. However, in 2021–2022, the concentration index for outpatient visits per person significantly decreased to −0.04, indicating that lower-income populations increasingly utilized outpatient services during this period. This trend may reflect improved access to healthcare among economically disadvantaged groups.

Table 4 presents the average annual percentage change in inequality indices for healthcare resource availability across regions. The highest increase is observed in the distribution of hospital beds, indicating growing inequality in their accessibility. Inequality in the availability of physicians remains relatively stable, while primary healthcare institutions show a slight change.

It should also be noted that there is a strong positive correlation between the number of hospital beds per 10,000 population and hospitalizations per 100 population (Pearson’s correlation = 0.73), suggesting that hospitalizations follow bed availability.

## 4. Discussion

This study analyzed the trends and degree of inequality in the distribution of healthcare resources and the utilization of medical services from 2002 to 2023 in the Republic of Kazakhstan. Cross-country evidence varies considerably across indicators, as most published studies report inequality measures for individual resources within specific national settings. Therefore, comparisons in this section are provided by indicator rather than by region, using data from representative studies in Asia, Africa, and OECD countries where available. The analysis of inequality in the distribution of primary care organizations per 10,000 population remained low-to-moderate over the study period (Gini = 0.15 in 2022). The distribution of hospital beds was low by this classification (Gini = 0.10 in 2023). These values suggest that, at the macro-regional level, resources are comparatively evenly distributed, although directional gradients persist.

Despite this, southern regions are systematically less supplied with facilities, beds, and physicians, yet record higher outpatient visits and hospitalizations, indicating a mismatch between need and supply. A comparable pattern—formally even facility distribution coexisting with utilization pressures in high-growth areas—has been documented in Saudi Arabia [29].

In terms of medical personnel availability in Kazakhstan, the numbers are comparable to other countries. In China, the distribution of doctors per 1000 population ranges from 1.96 to 2.78 in Western China and 0.46 in Shanghai, whereas in Kazakhstan, this figure ranges from 1.3 to 3.8 at the regional level and 4.2 in the capital. The availability of doctors in Kazakhstan is consistent with, and sometimes exceeds, these international standards [30]. Kazakhstan’s physician density falls within ranges reported internationally and is higher in the capital, consistent with tertiary-care and training-center concentration. Prior research also shows growth in rural health workers post-2009, though population-adjusted gains were modest, and substantial inter-regional gaps in rural nurses and physicians persist [31].

In Shanghai, China, inequality across resources has been reported in the moderate range (Gini ≈ 0.25–0.39) [32]. By contrast, Ethiopia shows high inequality for several indicators (e.g., Gini for physicians = 0.612), whereas our regional-level estimates for Kazakhstan remain in the low-to-moderate range (e.g., physicians ≈ 0.13; hospitalizations per capita ≈ 0.005) [33].

The availability of hospital beds is a critical indicator for assessing healthcare capacity, reflecting the distribution of resources such as human resources and medical equipment [34]. In China, the distribution shows relative inequality, with a national-level inequality index for hospital beds at 0.367 in 2019, although this can still be considered relatively equitable [35].

In India, the Gini coefficient for the distribution of doctors in 2013 was 0.21, with significant inequality noted in the distribution of specialists, surgeons, and pediatricians [36]. The Gini coefficient has some limitations. For instance, in Mongolia, the Gini coefficients for the distribution of doctors, nurses, and hospital beds by population share were 0.07 and 0.06, respectively. However, the Gini coefficients for the distribution of nurses and hospital beds by area were 0.67 and 0.69, respectively, highlighting the limitations of the Gini index in countries with large territories and low population density [37]. For Kazakhstan, this implies that low regional Gini values may still mask spatial access frictions in sparsely populated areas, underscoring the value of higher-resolution data.

Population density in Kazakhstan is unevenly distributed. According to the Ministry of National Economy, the southern regions have the highest population density, while the central and western regions are less populated. Consequently, the number of hospitals, primary healthcare organizations, and doctors per 10,000 population is lower in the southern regions. Notably, the concentration of doctors is higher in regions with medical universities [38]. US researchers noted a weak positive correlation between income inequality and inequality in the distribution of doctors and hospital beds [39]. Globally, doctors are more willing to work in major cities, particularly where medical universities are located, compared to rural and remote areas [40,41]. Another study, focused in United States and Japan, revealed that despite a continual rise in the number of physicians, their distribution does not align with population distribution [5]. In Korea, it was identified that the availability of MRI machines correlates with factors such as the proportion of the population aged over 65, regional income per capita, the presence of major hospitals, and university-affiliated hospitals [6].

Polish researchers analyzing inequality in the distribution of doctors concluded that medical personnel generally gravitate towards areas with the greatest need, and analyzing healthcare resource inequality can inform effective policy reforms [42]. In Kazakhstan, inequality in the distribution of doctors is observed between urban and rural healthcare organizations. In 2021, the availability of doctors in rural areas was 17.2 per 10,000 population compared to 57.1 per 10,000 population in urban areas. In 2017, this figure was 14.47 for rural areas and 43.71 per 10,000 population for urban areas, indicating a slight increase over five years but no significant changes. According to the Government of Kazakhstan, key factors contributing to the shortage of medical personnel include migration to neighboring countries, professional dissatisfaction, low salaries, difficult working conditions, increasing responsibilities, and a lack of career growth prospects [6].

The concentration index for the use of medical services relative to nominal income per capita at both inpatient and outpatient levels showed relative equality throughout the study period. In China, in 2014, the concentration index for the number of hospitalizations and outpatient visits was 0.2, with notable inequality in the wealthier eastern regions compared to the poorer western and central regions [43].

Similar results have been observed in other studies. In regions with poorer populations, outpatient medical service utilization is higher than inpatient services, whereas, in wealthier regions, the opposite is true. This is attributed to the disproportionate concentration of poor populations in rural areas and the disproportionate distribution of hospitals in urban areas [44]. In the most populous country, the concentration index for public inpatient care among the poor was 0.05 in 2018, compared to 0.09 among the wealthy, and 0.17 for private inpatient care among the wealthy, indicating a significant disparity [45]. Our study aligns with these findings, showing that inpatient services dominate in high-income regions. Similar results were found in studies conducted in Africa, Asia, and Latin America [46]. We also assume the presence of the Roemer’s Law effect, as identified in previous studies [47,48].

The temporary disruptions caused by COVID-19—including reduced PHC throughput and emergency hospital expansion were visible in 2020–2021 but did not alter the long-term spatial gradients of inequality identified in this study.

This study has certain limitations that should be acknowledged. Firstly, aggregated data were used for the analysis. Secondly, due to the lack of higher-resolution data, inequality indices were calculated at the regional level, although it would have been preferable to calculate them at the district level. As a result, the findings mainly capture macro-level structural disparities between regions and may underestimate local variations, including the rural–urban gap. Nevertheless, the regional approach remains informative for identifying long-term spatial gradients and national patterns, while future research using district- or facility-level data could provide a more detailed view of intra-regional inequality.

Our study’s strengths include the use of long-term official data, rigorous inequality measures, and clustering of time series to identify regional disparities. This approach provides valuable insights into healthcare system dynamics and regional inequalities.

## 5. Conclusions

This study examined long-term inequality in healthcare resources and service utilization across Kazakhstan during 2002–2023. Overall, the distribution of primary healthcare (PHC) facilities, hospital beds, and medical personnel remains relatively equitable, yet persistent regional disparities indicate that capacity does not always align with population need. Southern regions, for instance, combine higher service utilization with lower resource availability.

Strengthening PHC should therefore focus not only on infrastructure expansion but also on improving the balance between human resources, funding, and population demand. Regional planning mechanisms could be enhanced to ensure that new investments and workforce programs prioritize underserved areas. The experience of the COVID-19 pandemic, which temporarily shifted resources toward emergency hospital care, further highlights the importance of maintaining resilient PHC and equitable access in times of crisis.

Continued monitoring of inequality indicators and the integration of spatial analytics into policy planning can support data-driven decisions toward universal health coverage. By linking equity goals with regional healthcare development strategies, Kazakhstan can sustain progress toward a more balanced and responsive healthcare system.

## Figures and Tables

**Figure 1 ijerph-22-01762-f001:**
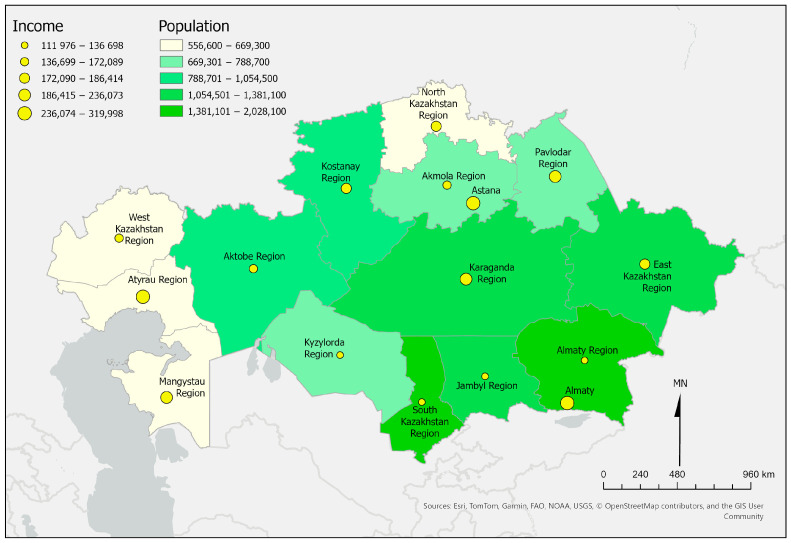
Regions of Kazakhstan by absolute income per person and population in 2023.

**Figure 2 ijerph-22-01762-f002:**
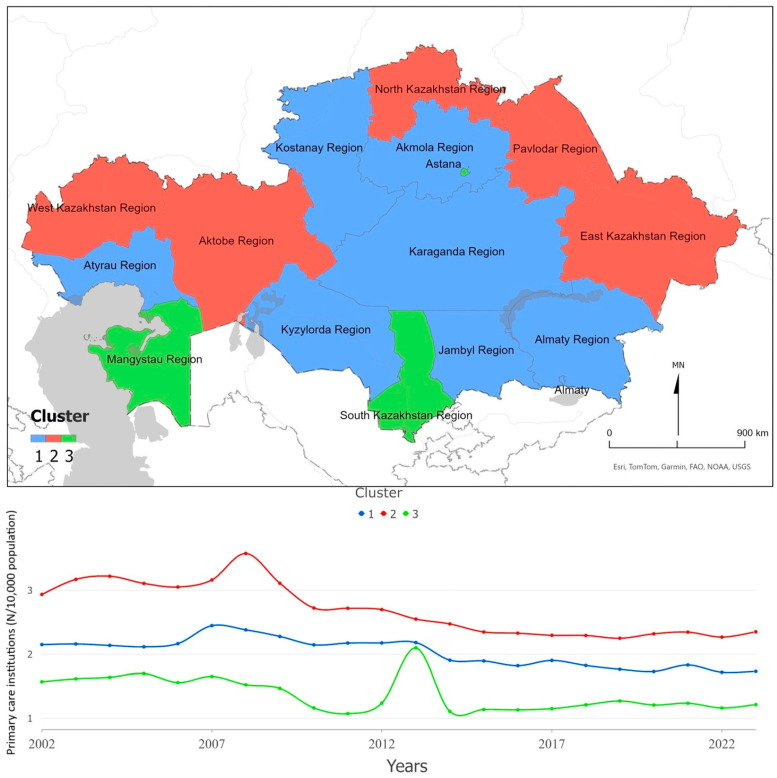
Primary care institutions per 10,000 population, Kazakhstan (2002–2023). The figure shows spatial clusters (**upper panel**) and the corresponding temporal trends (**lower panel**).

**Figure 3 ijerph-22-01762-f003:**
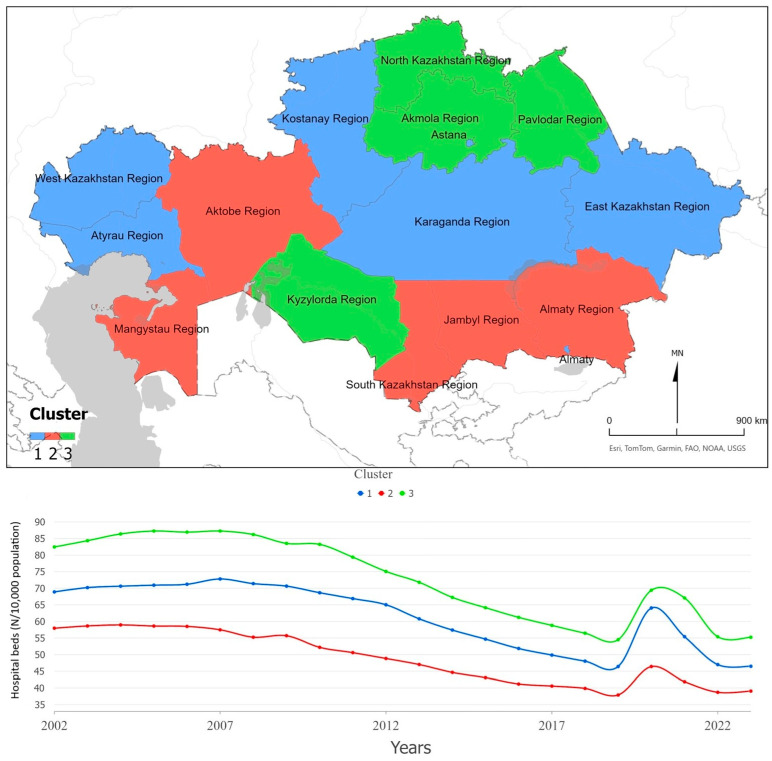
Hospital beds per 10,000 population, Kazakhstan (2002–2023). The figure shows spatial clusters (**upper panel**) and the corresponding temporal trends (**lower panel**).

**Figure 4 ijerph-22-01762-f004:**
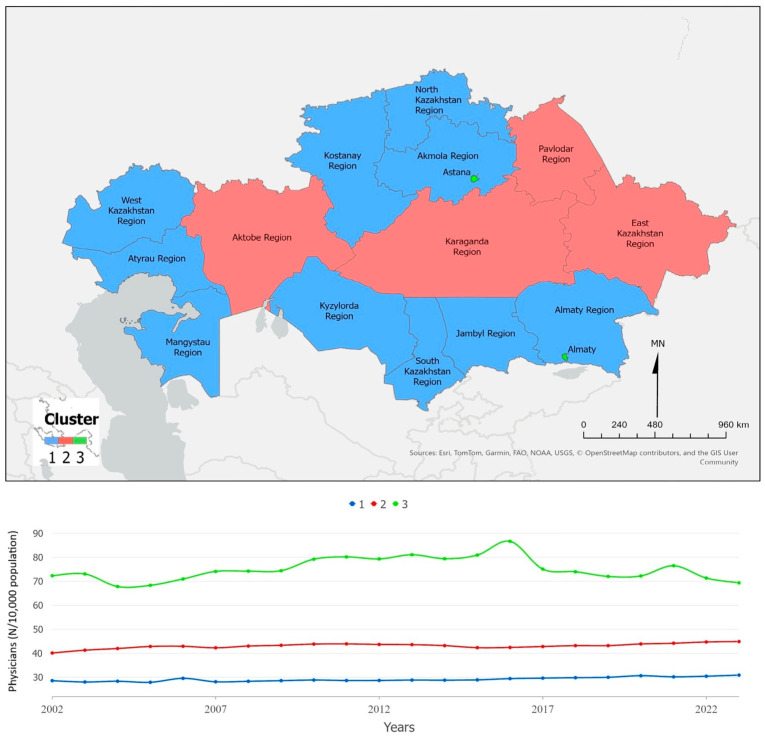
Physicians (excluding dentists) per 10,000 population, Kazakhstan (2002–2023). The figure shows spatial clusters (**upper panel**) and the corresponding temporal trends (**lower panel**).

**Figure 5 ijerph-22-01762-f005:**
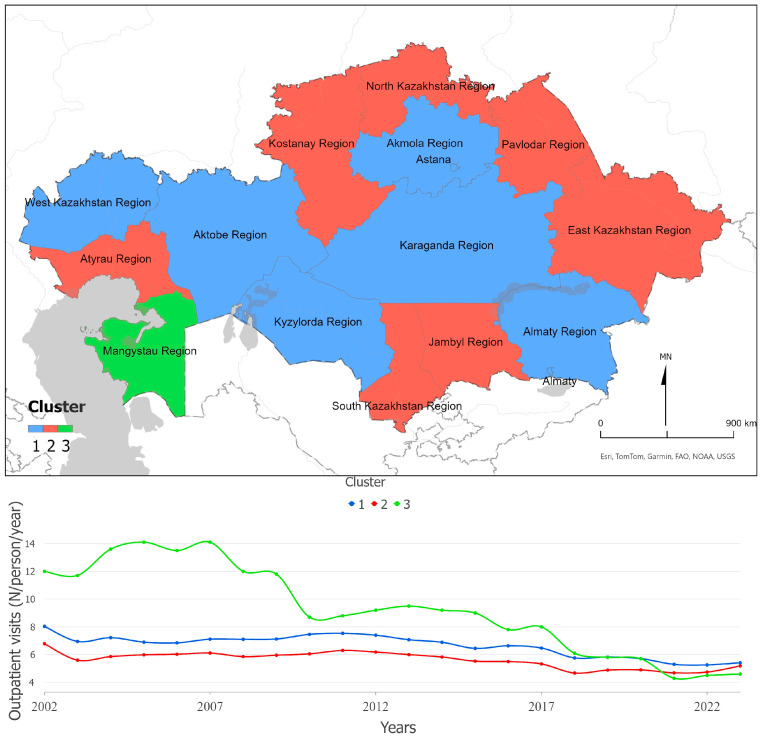
Outpatient visits per 100 population, Kazakhstan (2002–2023). The figure shows spatial clusters (**upper panel**) and the corresponding temporal trends (**lower panel**).

**Figure 6 ijerph-22-01762-f006:**
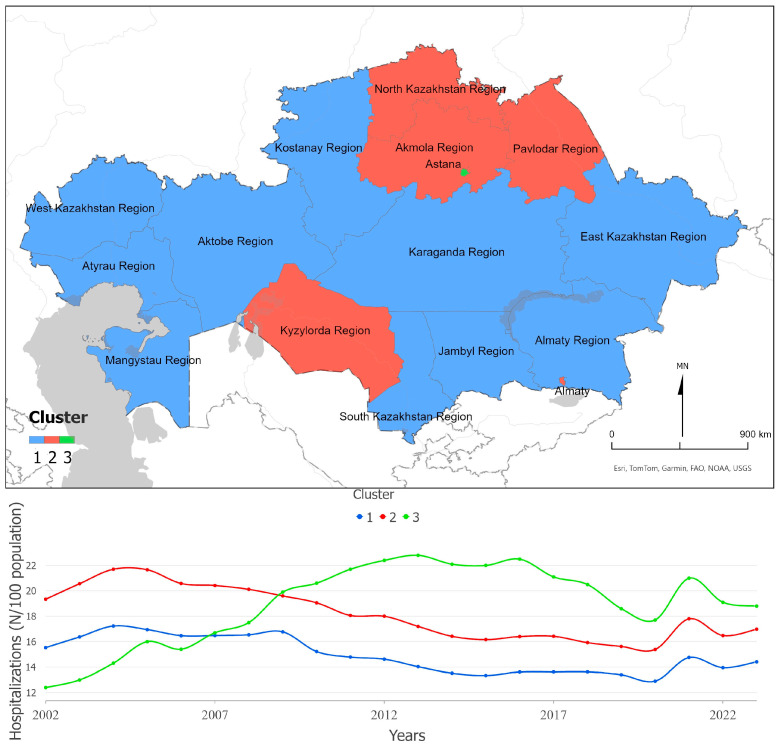
Hospitalizations per 100 population, Kazakhstan (2002–2023). The figure shows spatial clusters (**upper panel**) and the corresponding temporal trends (**lower panel**).

**Figure 7 ijerph-22-01762-f007:**
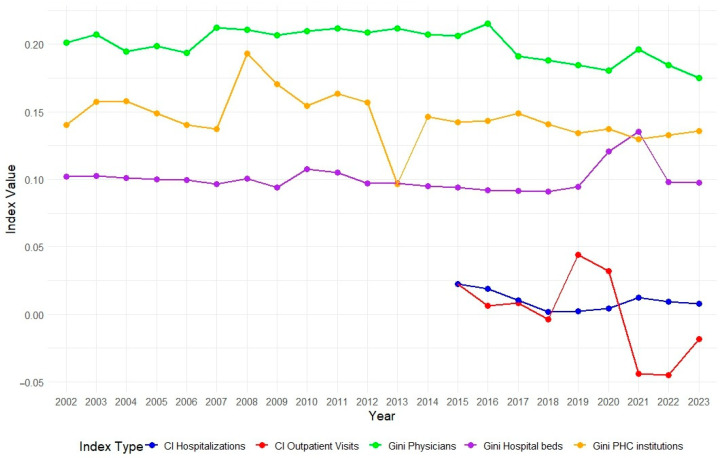
Dynamics of inequality indices (Gini and concentrarion) among Kazakhstan regions from 2002 to 2022.

**Table 1 ijerph-22-01762-t001:** Descriptive statistics of distribution of health resources in 2023.

Region	Primary Care Institutions per 10,000 Population	Hospital Beds per 10,000 Population	Outpatient Visits per Person per Year	Hospitalizations per 100 Population	Physicians per 10,000 Population	Per Capita Income in KZT *
Pavlodar Region	2.24	50.30	5.20	16.60	40.85	199,404
Jambyl Region	1.76	45.20	5.40	14.40	29.11	129,051
Kostanay Region	1.71	52.60	4.50	14.30	30.49	178,837
Mangystau Region	1.24	30.70	4.60	12.90	31.54	231,263
Karaganda Region	1.78	53.30	5.40	16.50	47.12	236,073
Kyzylorda Region	1.91	53.50	5.90	18.00	35.56	136,698
East Kazakhstan Region	2.21	49.90	5.20	17.00	42.86	186,414.5
Aktobe Region	2.46	39.20	5.60	14.10	48.67	163,509
Atyrau Region	1.86	35.80	5.60	12.90	30.10	319,998
South Kazakhstan Region	1.06	37.80	5.60	15.90	35.90	111,976.5
Akmola Region	1.74	54.90	4.20	15.80	26.31	168,301
Almaty Region	1.50	42.50	6.50	12.70	24.36	129,880
North Kazakhstan Region	2.88	62.20	4.80	18.50	32.61	178,495
Almaty	1.59	41.80	5.50	16.00	64.94	288,339
West Kazakhstan Region	1.95	48.60	5.40	13.40	33.01	172,089
Astana	1.34	43.40	4.80	18.80	73.74	272,742

* 1 US Dollar = 454.56 KZT (in 31 December 2023).

**Table 2 ijerph-22-01762-t002:** Aggregated Gini and concentration indices in the hospital and primary care sectors (2023).

Indicator	Gini Index (Std. Error)	*p*-Value
Primary care institutions per 10,000	0.14 (0.11)	<0.001
Hospital beds per 10,000	0.1 (0.005)	<0.001
Physicians per 10,000	0.18 (0.02)	<0.001

**Table 3 ijerph-22-01762-t003:** Aggregated concentration indices for healthcare utilization (2023).

Indicator	Concentration Index (Std. Error)	*p*-Value
Outpatient visits per person per year	−0.018 (0.02)	0.26
Hospitalizations per 100	0.007 (0.02)	0.69

**Table 4 ijerph-22-01762-t004:** Average annual percentage change for inequality indices (2002–2023).

Index	Relative Increase	95% CI for Average Annual Percentage Change
Gini physicians	0.06%	−0.52 to 0.41%
Gini hospital beds	1.32%	0.87 to 1.77%
Gini PHC institutions	0.24%	−0.65 to 1.13%

## Data Availability

Data available on request from the authors.

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
