# Peer review of "Inequality in the Distribution and Utilization of Healthcare Resources in Kazakhstan (2002–2023): A Spatiotemporal Analysis"

_ijerph, 2025, doi:10.3390/ijerph22111762_

Round 1
Reviewer 1 Report
Comments and Suggestions for Authors
The manuscript presents an important and timely study on inequality in healthcare resources and utilization in Kazakhstan during 2002–2023. The study uses long-term national data which make analysis more reliable and relevant for public health. The methodology employed Gini index, Concentration Index and GIS-based spatiotemporal cluster analysis. These methods are already used in other international studies, but their application in Kazakhstan context and Central Asia region is still limited, so the paper add new evidence for this setting.
However, the presentation and explanation of results are not always clear, and some descriptions are inconsistent with the figures. Some comments are provided below so that the authors take them into consideration.
- The introduction remains descriptive and not framed in theoretical concept. It explains background and mentions universal health coverage, but it does not provide framework that link socio-economic status, health resources, and utilization. Without this, the analysis looks mainly descriptive. Authors should consider using an established health system or health service utilization framework to connect socioeconomic factors, health resources, and utilization.
- The novelty is not well explained. Spatiotemporal analysis has been used in other countries, so authors should clarify what is new here. This will show why the study is valuable for international reader, not only for local interest.
- The analysis uses only regional level (oblast and two big cities). This approach is understandable due to data availability, but it misses important inequalities inside region. For example, rural–urban gap or difference between districts are not seen in this paper. The limitation is mentioned but very briefly. Authors should discuss more how this limitation affect the interpretation and what future study can address.
- Table 1 is presented with many numbers for 2023. However, in the text, authors only say general information is shown. This is not useful for reader. A table should not only list numbers but highlight the important disparity. The main message is not extracted. Authors should highlight which regions are highest, lowest, and how income relates with distribution of physicians or hospital beds.
- In Figure 2, there is inconsistency. The text describes lowest values over 20 years are South Kazakhstan, West Kazakhstan, and Astana. But in the figure, the green cluster of lowest is actually South Kazakhstan, Mangystau, and Astana. West Kazakhstan appears in red cluster, not lowest. This mismatch can confuse reader.
- Also, spelling is inconsistent (Mangystau in figure, Mangistau in text, line 226).
- In Figure 4, similar inconsistency exist. In line 216, the manuscript describes the red cluster as Aktobe, Karaganda, Pavlodar, and East Kazakhstan. But the figure show red cluster is West Kazakhstan, Aktobe, North Kazakhstan, Pavlodar, East Kazakhstan. Karaganda is not red cluster, it is blue. Such error reduce credibility of results presentation. The authors need to carefully revise description and make consistency between figure and text.
- The results section present Mann–Kendall test values and p-values, but do not explain what they mean. A negative significant value mean decreasing trend, positive mean increasing trend, and non-significant mean stable. Authors should write short interpretation in words, not only numbers. In the discussion, the meaning of these trends should be linked to health system or policy factors, otherwise the test adds little value.
- In Discussion, the manuscript compare Gini coefficients with several countries, but the comparison is presented mainly as numbers and give impression of better or worse. It would be more useful if authors frame the comparison as context, to explain what level of inequality the Kazakhstan values represent (low, moderate, or high) and what this mean for policy relevance. This way the discussion becomes less about ranking countries and more about understanding the practical meaning of inequality values.
- COVID-19 is only mentioned briefly in Results and Conclusion, but without deeper analysis. It would strengthen the paper if authors discuss how the pandemic affected the distribution and utilization of resources, for example through reduction of PHC capacity, emergency allocation of hospital beds, or shift in utilization pattern. This will make the interpretation of recent years more meaningful.
- The conclusion remains too general. It repeats the results and says PHC should be strengthened, but it is not clear what exact policy action is needed. Authors should give more concrete recommendation. This will improve the practical usefulness of the study for policy.
Minor issues
- line 23 the word Global is written in bold, but it should be normal text. Please check formatting.
- line 99 there is line break in the middle of a sentence. This should be corrected for readability.
- Figure 1 uses green background with red letters, which is hard to read. Authors should improve resolution or adjust color so that names are visible.
The English is overall understandable, however, there are minor inconsistencies in spelling, formatting, and clarity. Authors should revise the text for consistency and clarity.
Author Response
We sincerely thank the reviewer for the thorough and constructive feedback.
We carefully reviewed all comments and revised the manuscript accordingly to improve its clarity, analytical depth, and consistency between text, figures, and tables.
Where full implementation was not feasible due to data or methodological constraints, we provided clear explanations and clarifying notes in the text.
Overall, all feasible suggestions have been incorporated, and we believe that the revised version has substantially improved in quality and readability thanks to the reviewer’s valuable input.
Comment 1:
The introduction remains descriptive and not framed in theoretical concept. It explains background and mentions universal health coverage, but it does not provide framework that link socio-economic status, health resources, and utilization. Without this, the analysis looks mainly descriptive. Authors should consider using an established health system or health service utilization framework to connect socioeconomic factors, health resources, and utilization.
Response:
We agree with the reviewer that the Introduction should more clearly articulate the conceptual pathway linking socioeconomic conditions, healthcare resources, and service utilization. We have revised the Introduction to explicitly describe this linkage, drawing on established frameworks such as the Andersen Behavioral Model of Health Services Use and the WHO health system perspective. This addition clarifies that inequity in utilization is not only a function of population need but also of resource distribution, system capacity, and financial accessibility.
Changes in manuscript:
Added to the Introduction:
«Health service utilization is shaped by both demand-side and supply-side determinants. Following the Andersen Behavioral Model of Health Services Use, socioeconomic context (for example, income) influences an individual’s ability to seek care, while predisposing and enabling factors determine whether care is actually obtained [13]. At the same time, in line with the WHO health system perspective, the distribution of infrastructure (primary care facilities, hospital beds), workforce (medical personnel), and financial protection mechanisms are system-level inputs that determine service availability and capacity at the regional level [14]. Inequality in utilization can therefore be interpreted as the combined result of unequal resource allocation, uneven system capacity, and differences in local population needs. In this study, we examine these dimensions jointly by assessing regional disparities in availability (infrastructure and personnel) and in actual use of care».
Comment 2:
The novelty is not well explained. Spatiotemporal analysis has been used in other countries, so authors should clarify what is new here. This will show why the study is valuable for international reader, not only for local interest.
Response:
We appreciate the reviewer’s observation. We clarified in the Introduction that our study does not introduce new methods but applies established inequality and spatial analysis tools to a setting that has not been previously examined. This is the first national-level assessment in Kazakhstan and the broader Central Asian region using long-term regional data (2002–2023) to analyze inequality in healthcare resources and service utilization. The approach provides an initial macro-level understanding of structural disparities based on secondary data.
Changes in manuscript:
While spatiotemporal clustering methods have been increasingly applied in public health internationally, and standardized approaches to measuring health inequality are well established, there remains no long-term, country-wide analysis that combines these methods for Kazakhstan or the Central Asian region. The present study addresses this gap by integrating inequality indices with spatiotemporal clustering over 2002-2023 [17,18].
Comment 3:
The analysis uses only regional level (oblast and two big cities). This approach is understandable due to data availability, but it misses important inequalities inside region. For example, rural–urban gap or difference between districts are not seen in this paper. The limitation is mentioned but very briefly. Authors should discuss more how this limitation affect the interpretation and what future study can address.
Response:
We fully agree with the reviewer that using aggregated regional (oblast) data limits the ability to capture intra-regional or rural–urban differences. Our study was based on the structure of publicly available national statistics, which provide consistent time series only at the regional level.
To address this concern, we have expanded the Limitations section. The revised text now clarifies that our findings describe macro-level structural inequalities between regions rather than within them. We also emphasize that this approach still allows identification of persistent geographic gradients. In future studies, we plan to complement this macro-level perspective with district or facility-level data, which would permit analysis of local accessibility and the rural–urban gap.
Changes in manuscript in the limitation section:
This study has certain limitations that should be acknowledged. Firstly, aggregated data were used for the analysis. Secondly, due to the lack of higher-resolution data, inequality indices were calculated at the regional level, although it would have been preferable to calculate them at the district level. As a result, the findings mainly capture macro-level structural disparities between regions and may underestimate local variations, including the rural–urban gap. Nevertheless, the regional approach remains informative for identifying long-term spatial gradients and national patterns, while future research using district- or facility-level data could provide a more detailed view of intra-regional inequality.
Comment 4:
Table 1 is presented with many numbers for 2023. However, in the text, authors only say general information is shown. This is not useful for reader. A table should not only list numbers but highlight the important disparity. The main message is not extracted. Authors should highlight which regions are highest, lowest, and how income relates with distribution of physicians or hospital beds.
Response:
We thank the reviewer for this valuable comment. We agree that the description accompanying Table 1 should emphasize the main patterns rather than restate raw numbers. In the revised Results section, we have added a concise summary that highlights the key disparities — identifying regions with the highest and lowest resource availability and linking these with income levels and geographic gradients.
Changes in manuscript:
Added to Results, before Table 1:
“In 2023, physician density ranged from 26.3 per 10 000 population in Akmola to 48.7 in Aktobe, while hospital-bed supply was lowest in Mangystau (30.7) and highest in North Kazakhstan (62.2). Regions with higher income, such as Atyrau, Astana, and Almaty, consistently showed greater healthcare resource availability, illustrating a persistent north–south gradient.”
Comment 5:
In Figure 2, there is inconsistency. The text describes lowest values over 20 years are South Kazakhstan, West Kazakhstan, and Astana. But in the figure, the green cluster of lowest is actually South Kazakhstan, Mangystau, and Astana. West Kazakhstan appears in red cluster, not lowest. This mismatch can confuse reader.
Response:
We thank the reviewer for noticing this inconsistency. The text in the Results section has been corrected to match the actual cluster composition shown in Figure 2. It now specifies that the lowest values over the study period were observed in South Kazakhstan, Mangystau, and Astana, which correspond to the green cluster in the figure.
Changes in manuscript:
All figure captions and related text in the Results section (Figures 2–6) were revised to match the spatial–temporal clusters and to clearly indicate which colors correspond to each group of regions.
Comment 6:
Also, spelling is inconsistent (Mangystau in figure, Mangistau in text, line 226).
Response:
We thank the reviewer for pointing this out. The spelling inconsistency has been corrected throughout the manuscript. The name “Mangystau” is now used consistently in both the text and all figures.
Comment 7:
In Figure 4, similar inconsistency exists. In line 216, the manuscript describes the red cluster as Aktobe, Karaganda, Pavlodar, and East Kazakhstan. But the figure show red cluster is West Kazakhstan, Aktobe, North Kazakhstan, Pavlodar, East Kazakhstan. Karaganda is not red cluster, it is blue. Such error reduces credibility of results presentation. The authors need to carefully revise description and make consistency between figure and text.
Response:
Thank you for this careful observation. The inconsistency was due to an outdated version of Figure 4 being used in the draft, which did not correspond to the final text description. We have corrected this issue by updating Figure 4 with the correct spatial–temporal clustering output, and we revised the accompanying paragraph in the Results section accordingly.
In the revised manuscript, the description of clusters (including which regions form the higher-staffed and lower-staffed groups) now matches the finalized version of Figure 4.
Comment 8:
The results section presents Mann–Kendall test values and p-values, but do not explain what they mean. A negative significant value means decreasing trend, positive mean increasing trend, and non-significant mean stable. Authors should write short interpretation in words, not only numbers. In the discussion, the meaning of these trends should be linked to health system or policy factors, otherwise the test adds little value.
Response:
We appreciate the reviewer’s insightful comment. All descriptions accompanying Figures 2–6 in the Results section have been revised to include brief interpretations of the Mann–Kendall trend test results. For each indicator, we now specify whether the trend is increasing, decreasing, or stable, and we provide short contextual explanations linking these trends.
Comment 9:
In Discussion, the manuscript compare Gini coefficients with several countries, but the comparison is presented mainly as numbers and give impression of better or worse. It would be more useful if authors frame the comparison as context, to explain what level of inequality the Kazakhstan values represent (low, moderate, or high) and what this mean for policy relevance. This way the discussion becomes less about ranking countries and more about understanding the practical meaning of inequality values.
Response:
Thank you for this valuable suggestion. We revised the Discussion to shift from numeric listings to contextual interpretation:
Changes in manuscript :
Cross-country evidence varies considerably across indicators, as most published studies report inequality measures for individual resources within specific national settings. Therefore, comparisons in this section are provided by indicator rather than by region, using data from representative studies in Asia, Africa, and OECD countries where available. The analysis of inequality in the distribution of primary care organizations per 10 000 population remained low-to-moderate over the study period (Gini = 0.15 in 2022). The distribution of hospital beds was low by this classification (Gini = 0.10 in 2023). These values suggest that, at the macro-regional level, resources are comparatively evenly distributed, although directional gradients persist.
Despite this, southern regions are systematically less supplied with facilities, beds, and physicians, yet record higher outpatient visits and hospitalizations, indicating a mismatch between need and supply. A comparable pattern—formally even facility distribution coexisting with utilization pressures in high-growth areas—has been documented in Saudi Arabia [29].
Kazakhstan’s physician density falls within ranges reported internationally and is higher in the capital, consistent with tertiary-care and training-center concentration. Prior research also shows growth in rural health workers post-2009, though population-adjusted gains were modest, and substantial inter-regional gaps in rural nurses and physicians persist [31].
In Shanghai, China, inequality across resources has been reported in the moderate range (Gini ≈ 0.25–0.39) [32]. By contrast, Ethiopia shows high inequality for several indicators (e.g., Gini for physicians = 0.612), whereas our regional-level estimates for Kazakhstan remain in the low-to-moderate range (e.g., physicians ≈ 0.13; hospitalizations per capita ≈ 0.005) [33].
For Kazakhstan, this implies that low regional Gini values may still mask spatial access frictions in sparsely populated areas, underscoring the value of higher-resolution data.
Comment 10:
COVID-19 is only mentioned briefly in Results and Conclusion, but without deeper analysis. It would strengthen the paper if authors discuss how the pandemic affected the distribution and utilization of resources, for example through reduction of PHC capacity, emergency allocation of hospital beds, or shift in utilization pattern. This will make the interpretation of recent years more meaningful.
Response:
We appreciate this thoughtful comment. In the revised version, we expanded the Discussion section to briefly explain how the COVID-19 pandemic temporarily affected the spatial and temporal patterns of healthcare resource distribution and utilization. The new text notes that the pandemic led to a short-term expansion of hospital capacity and a reduction in PHC throughput, reflecting emergency reallocation of resources and changes in care-seeking behavior. We intentionally kept the addition concise, as the focus of this study remains on long-term inequality rather than pandemic-specific outcomes.
Changes in manuscript:
The temporary disruptions caused by COVID-19 - including reduced PHC throughput and emergency hospital expansion were visible in 2020 - 2021 but did not alter the long-term spatial gradients of inequality identified in this study.
Comment 11:
The conclusion remains too general. It repeats the results and says PHC should be strengthened, but it is not clear what exact policy action is needed. Authors should give more concrete recommendation. This will improve the practical usefulness of the study for policy.
Response:
We appreciate this insightful comment. The Conclusion section has been revised to make it more specific and policy-oriented while keeping it concise and consistent with the overall structure of the manuscript. The revised version no longer repeats the numerical results but instead focuses on the implications for health-system planning and policy.
It now emphasizes that while inequality in Kazakhstan remains low-to-moderate, regional disparities persist—especially the under-supply of resources in southern regions and concentration of workforce in urban centers. The updated text highlights the need to strengthen PHC not only through infrastructure expansion but also by improving human-resource distribution, alignment of funding with population needs, and coordination between PHC and hospital care.
We also added a brief reference to COVID-19’s temporary impact on service distribution and to the importance of monitoring inequality indicators for data-driven decision-making.

Reviewer 2 Report
Comments and Suggestions for Authors
The manuscript reads as largely descriptive: data presentation dominates, while a clearly stated research problem and testable hypotheses are missing. This makes it harder to assess the paper’s contribution. Main comments, key improvements needed:
1.Absence of a thesis and research questions. The authors describe what was computed or observed but do not formulate a problem or hypotheses to test.
2.Introduction, line 62: it is unclear whether Kazakhstan has universal health insurance. Please specify the status and scope, and cite a source.
3.Line 86: the literature gap is not identified, nor is it explained how the paper addresses it.
4.Table 1 (descriptive statistics, 2023) adds little beyond the raw dataset.
Consider moving it to the supplement or replacing it with visualizations that support the paper’s arguments.
5.Figures: placing two figures side‑by‑side reduces readability - please separate them and standardize the captions.
6.Results are presented in a report‑like, descriptive manner.
Instead of answering the research questions, the section lists indicators.
7.Discussion: the most valuable element is the attempt to compare the Gini coefficient between countries; it is worth organizing the comparisons (e.g. first Eastern Europe, then Central Asia, then OECD) and linking them to hypotheses.
Author Response
We sincerely thank the reviewer for the thorough and constructive feedback.
We carefully reviewed all comments and revised the manuscript accordingly to improve its clarity, analytical depth, and consistency between text, figures, and tables.
Where full implementation was not feasible due to data or methodological constraints, we provided clear explanations and clarifying notes in the text.
Overall, all feasible suggestions have been incorporated, and we believe that the revised version has substantially improved in quality and readability thanks to the reviewer’s valuable input.
Comment 1:
Absence of a thesis and research questions. The authors describe what was computed or observed but do not formulate a problem or hypotheses to test.
Response:
We appreciate this valuable observation. In the revised version, we have clearly formulated the research problem and hypothesis.
Changes in manuscript:
Added to the end of the Introduction:
“The aim of this study was to assess regional inequality in the distribution and utilization of healthcare resources across Kazakhstan during 2002-2023, using Gini and concentration indices and spatiotemporal cluster analysis to identify patterns of disparity among regions.”
Comment 2:
Introduction, line 62: it is unclear whether Kazakhstan has universal health insurance. Please specify the status and scope, and cite a source.
Response:
We appreciate the reviewer’s observation. To clarify this distinction, we added one sentence explaining that Kazakhstan introduced a compulsory social health insurance (SHI) system in 2020, which functions as a key mechanism to achieve UHC. This amendment ensures terminological precision and addresses the reviewer’s concern.
Changes in manuscript:
Added to the end of the paragraph beginning “Kazakhstan is committed to achieving universal healthcare coverage…”:
“Since 1 January 2020, Kazakhstan has operated a compulsory Social Health Insurance (SHI) system alongside the State Guaranteed Benefits Package, providing healthcare coverage for all citizens and permanent residents as part of progress toward Universal Health Coverage (UHC) [Law No. 405-V ‘On compulsory social medical insurance’, 2015].”
Reference added:
Law of the Republic of Kazakhstan No. 405-V On Compulsory Social Medical Insurance (16 November 2015). Available at: https://adilet.zan.kz/eng/docs/Z1500000405
Comment 3:
Line 86: the literature gap is not identified, nor is it explained how the paper addresses it.
Response:
We appreciate this comment. We revised the end of the Introduction to explicitly articulate the research gap. We now state that although spatiotemporal clustering has been applied in international public-health research and standardized approaches to health-inequality measurement are well established, no long-term, country-wide study combining these methods exists for Kazakhstan or Central Asia. We then specify that our study addresses this gap by integrating inequality indices with spatiotemporal clustering for 2002–2023.
Changes in manuscript:
Added two sentences before the aim statement in the Introduction
Comment 4:
Table 1 (descriptive statistics, 2023) adds little beyond the raw dataset. Consider moving it to the supplement or replacing it with visualizations that support the paper’s arguments.
Response:
We appreciate the reviewer’s suggestion. We retained Table 1 in the main text because it provides essential context on regional healthcare resources and income in 2023. To enhance its analytical value, we added interpretive text highlighting key disparities and relationships with income and regional gradients.
Changes in manuscript:
Added to Results, before Table 1:
“In 2023, physician density ranged from 26.3 per 10 000 population in Akmola to 48.7 in Aktobe, while hospital-bed supply was lowest in Mangystau (30.7) and highest in North Kazakhstan (62.2). Regions with higher income, such as Atyrau, Astana, and Almaty, consistently showed greater healthcare resource availability, illustrating a persistent north–south gradient.”
Comment 5:
Figures: placing two figures side by side reduces readability – please separate them and standardize the captions.
Response:
We appreciate the reviewer’s suggestion. In the revised version, we clarified that each figure represents a single analytical output generated by the Space–Time Cube and Time Series Clustering tools in ArcGIS Pro. Each output consists of two components — the spatial distribution of clusters (map, upper panel) and the corresponding temporal trend (chart, lower panel). These are produced together as a single block in the software, and therefore we retained them as integrated figures.
All captions were standardized to a uniform format that clearly specifies the indicator, study period, and the analytical method, ensuring consistency and readability across the manuscript.
Changes in manuscript:
Unified figure captions (Figures 2–6) following the same format:
“Figure X. [Indicator], Kazakhstan (2002–2023). The figure shows spatial clusters (upper panel) and the corresponding temporal trends (lower panel).”
The layout (map above, chart below) was retained to reflect the native output of the ArcGIS analytical workflow.
Comment 6:
“Results are presented in a report-like, descriptive manner. Instead of answering the research questions, the section lists indicators.”
Response:
We sincerely thank the reviewer for this insightful comment. In the revised manuscript, the Results section has been substantially rewritten to move beyond descriptive reporting and to provide analytical interpretation that directly addresses the study aim.
Specifically, we added explanatory links between the observed spatial–temporal patterns and national health system dynamics. Each figure (2-6) now includes a concise analytical paragraph that interprets the direction and meaning of the Mann–Kendall trends, connects cluster patterns to regional characteristics, and highlights policy or structural factors underlying the observed disparities.
Changes in manuscript:
Descriptions of figures (2-6) were changed.
Comment 7:
“Discussion: the most valuable element is the attempt to compare the Gini coefficient between countries; it is worth organizing the comparisons (e.g., first Eastern Europe, then Central Asia, then OECD) and linking them to hypotheses.”
Response:
We thank the reviewer for this valuable suggestion. We fully acknowledge the importance of structuring international comparisons in a clear and systematic way. However, the available literature on healthcare inequality provides comparable Gini or Concentration Indices only for specific indicators (such as physicians, hospital beds, or PHC facilities) within individual countries, rather than for all indicators across all regions (OECD, Africa, Asia, etc.).
Therefore, to ensure analytical consistency, the Discussion remains structured by indicator rather than by region. This approach allows for meaningful cross-country comparison using comparable measures while avoiding misleading generalizations.
Changes in manuscript:
We have added one clarifying sentence at the beginning of the Discussion to make this rationale explicit:
“Cross-country evidence varies considerably across indicators, as most published studies report inequality measures for individual resources within specific national settings. Therefore, comparisons in this section are provided by indicator rather than by region, using data from representative studies in Asia, Africa, and OECD countries where available.”

Round 2
Reviewer 1 Report
Comments and Suggestions for Authors
Thank you for the thorough revision. The authors have satisfactorily addressed all of my major comments, and the manuscript has improved in clarity and consistency.